# Hepatocellular Carcinoma: The Role of Immunotherapy and Transplantation in the Era of Transplant Oncology

**DOI:** 10.3390/cancers15215115

**Published:** 2023-10-24

**Authors:** Saad Alghamdi, Waleed Al-Hamoudi

**Affiliations:** 1Liver & Small Bowel Health Centre Department, Organ Transplant Center of Excellence, King Faisal Specialist Hospital & Research Center, Riyadh 11211, Saudi Arabia; walhamoudi@gmail.com; 2Liver Disease Research Center, College of Medicine, King Saud University, Riyadh 11211, Saudi Arabia

**Keywords:** immunotherapy, liver transplantation, graft rejection, transplant oncology

## Abstract

**Simple Summary:**

Hepatocellular carcinoma (HCC) is a common cause of cancer-related deaths worldwide. During the early stages of the disease, HCC can be treated with surgery or radiofrequency procedures. Most HCC cases are discovered at later stages when these therapies cannot be used, and a treatment such as a liver transplant is needed. Recently, new options for the treatment of advanced HCC are available, called immune checkpoint inhibitors (ICIs). ICIs have been found to be safe and effective. However, there is concern that liver transplant patients may face graft rejection in both the pre- and post-transplant settings. Our review found that ICIs may be useful, especially in the pretransplantation setting. More data are needed to carefully select patients who will benefit from ICI treatment in both settings so that they can benefit from it while reducing harm.

**Abstract:**

Hepatocellular carcinoma (HCC) is one of the most common causes of cancer deaths worldwide. As most patients present with advanced disease, curative therapy such as surgical resection and radiofrequency ablation are rarely utilized. With the advent of immunotherapy, historical treatment approaches such as liver transplantation are being challenged. In particular, the use of immune checkpoint inhibitors (ICIs) has emerged as a safe and useful option in the treatment of HCC. However, there is concern over adverse effects, such as graft rejection and graft loss. This updated review discusses the role of immunotherapy in the pre- and post-transplantation setting and provides insights into the potential of immunotherapy as an adjunct to liver transplantation. We deliberate on the use of ICI in the setting of the Milan criteria as well as the University of California San Francisco’s expanded criteria for liver transplantation. Current data suggest that ICI has utility, especially in the pretransplantation setting. Nevertheless, larger, purposefully designed clinical trials are needed to clearly identify patients who will benefit most from ICI treatment in the transplant setting and determine parameters that will minimize the risk of graft rejection and maximize the benefits of this adjunct treatment.

## 1. Introduction

Hepatocellular carcinoma (HCC) is the most common primary liver cancer and was the third most common cause of cancer deaths worldwide in 2020 [1]. Surgical resection is a curative therapy option in patients with well-compensated liver function, as well as radiofrequency ablation in small tumors. However, a large percentage of patients with HCC present with cirrhotic disease, where liver transplantation remains the optimal management.

Historically, since 2007, sorafenib, a tyrosine kinase inhibitor (TKI), was the first and only US Food and Drug Administration-approved systemic therapy for advanced HCC [2]. Sorafenib was shown in randomized controlled trials (RCTs) to demonstrate a survival benefit versus placebo [3,4]. Following the introduction of sorafenib, it was not until recent years that newer systemic therapy options for advanced HCC became available. Since 2017, newer agents have been introduced, including regorafenib, cabozantinib, pembrolizumab, and ramucirumab in refractory disease, and lenvatinib and atezolizumab/bevacizumab in the first-line setting [5,6,7,8,9,10]. Current systemic therapies for advanced HCC include molecular targeted therapy (mainly TKIs and/or monoclonal antibodies), immune checkpoint inhibitors (ICIs), or a combination of both. 

Immunotherapy agents such as pembrolizumab and atezolizumab/bevacizumab have emerged as effective and safe options in the treatment of HCC, such that the latter is now offered as first-line treatment for most patients with advanced HCC, Child–Pugh class A, and Eastern Cooperative Oncology Group performance status 0–1 [11]. With the advent of immunotherapy, the role of historical treatment approaches such as liver transplantation is being challenged. This review discusses the role of immunotherapy in the pre- and post-transplantation setting and provides insights into the potential of immunotherapy as an adjunct to liver transplantation.

## 2. Outcomes in Post-immunotherapy Transplantation—Immunotherapy before Transplant

Liver transplantation has been one of the major treatment options for patients with HCC ever since the establishment of the Milan criteria in 1996 [12], which set out the eligibility of patients with HCC for liver transplantation. The Milan criteria state that tumors are amenable for transplantation if the tumor diameter of a single lesion is less than or equal to 5 cm or, for multiple lesions, no more than three tumors, each less than or equal to 3 cm, without vascular invasion or extrahepatic metastases. Liver transplantation is able to successfully treat HCC, producing 5-year overall survival rates of 60–85% [12,13,14]. However, in the real world, only a small fraction of patients have tumors that satisfy standard Milan criteria to receive liver transplantation. This is mainly due to the advanced stage that most patients present at, combined with a scarcity of neoadjuvant therapy to successfully downstage or delay tumor growth in patients waiting for a liver transplant. Patients who have tumors that do not fit the Milan criteria are usually downstaged using locoregional therapy; this approach not only reduces the risk of dropping off from the transplant waiting list but also decreases tumor dimensions so that they meet the acceptable criteria for liver transplantation.

The success in downstaging with locoregional therapy and the promising results from immunotherapy trials in advanced HCC have led to oncology specialists using immunotherapy as a downstaging strategy. However, there is a paucity of data supporting systemic therapy in the neoadjuvant setting and as a bridging strategy to liver transplantation. While there is a logical rationale behind using immunotherapy bridging therapy, as evidenced by the dropout rate of 10–20% in transplant waiting lists [15], unfortunately, there are no randomized control trials that systematically assess the role of immunotherapy-based bridging therapy on liver transplant outcomes. Some evidence has come from “accidental neoadjuvant” therapy, where immunotherapy was given as the destination therapy in patients with advanced disease who were not initially eligible for liver transplantation but were transitioned to the transplant pathway after achieving dramatic clinical responses [16].

Most of the data supporting the role of immunotherapy prior to transplantation come from case reports. Despite this limitation, bridging immunotherapy has been shown to have high efficacy in downstaging patients into the Milan criteria, thereby making them eligible for transplant. In general, programmed death-ligand 1 (PD-L1) receptor blockers produce results within 3 months of initiation and continue to be efficacious for some time, even after immunotherapy is withdrawn. This prolonged effect is explained by an extended half-life of the drugs and a prolonged duration of T-cell activation [17]. 

In the largest case series to date, Tabrizian et al. described nine patients who had recurrent HCC following liver resection as a primary treatment. These patients were successfully transplanted after receiving nivolumab as a bridging therapy. One third of the tumors demonstrated nearly full regression (>80%) on explant histology [18]. Qiao et al. reported a cohort of seven transplant recipients who received neoadjuvant pembrolizumab or camrelizumab plus lenvatinib. The objective response rate was 71%, according to the modified Response Evaluation Criteria in Solid Tumors (mRECIST) criteria. Only one patient suffered from mild acute rejection after transplant, but his liver function was restored after his immunosuppression regimen was adjusted [19]. Schwacha-Eipper et al. reported a patient with compensated cirrhosis who had recurrence after undergoing liver resection and progressed after sorafenib. He subsequently had a successful liver transplant after 34 cycles of nivolumab, with no evidence of allograft rejection [20]. A recent retrospective case review of 16 patients from China found complete or partial remission in the majority of patients (93.7%), although there was a 25% tumor recurrence rate at 1-year post-transplantation [21]. Several other studies have shown successful liver transplants at 12 months, with or without rejection that was resolved [22,23,24,25], while others have shown unsuccessful liver transplants due to fatal hepatic necrosis [26,27]. In addition, a recent case study in a pediatric patient reported the feasibility of using anti-PD1 therapy prior to orthotopic liver transplantation, where there were no signs of recurrent disease or any episode of rejection 48 months post orthotopic liver transplantation [28]. 

Besides the Milan criteria, the University of California San Francisco (UCSF) proposed an expanded set of criteria for liver transplantation. The UCSF criteria advocates for a downstaging pathway that allows patients slightly outside of the Milan criteria to be eligible for liver transplantation. It allows patients with a solitary tumor smaller than 6.5 cm or patients having three or fewer nodules, with the largest lesion being smaller than 4.5 cm or having a total tumor diameter less than 8.5 cm without vascular invasion, to undergo orthotopic liver transplantation. Based on this set of criteria in patient selection for liver transplantation, patients with HCC were found to have good survival rates of 75.2% at 5 years [29], suggesting that expanding the limits beyond the Milan criteria may benefit a wider set of patients. In terms of immunotherapy in the pretransplantation setting using the UCSF criteria, some success has been shown. In the dataset by Tabrizian et al., three out of the nine patients who received nivolumab pretransplantation were out of the Milan criteria but within the UCSF criteria. All three patients did not experience graft rejection after transplant [18]. In the recent case series of 16 patients in China, four exceeded UCSF criteria at diagnoses and were downstaged to UCSF criteria following ICI treatment. Two patients had acute graft rejection after liver transplantation, while two did not. Two had tumor recurrence post-transplantation (one in a patient with acute graft rejection and one who did not). Of note, the patient who had both graft rejection and tumor recurrence after liver transplantation had high alpha-fetoprotein levels pretransplant and had not received targeted therapy, indicating a higher tumor burden [21]. Interestingly, another case study of a patient with compensated cirrhosis secondary to hepatitis C virus and advanced HCC that was outside both the Milan and UCSF criteria successfully underwent liver transplantation after nivolumab treatment [25]. These results suggest that downstaging with an ICI in tumors within the USCF criteria is possible, although more data from carefully selected patient populations are needed to confidently use this strategy in these patients.

Transarterial chemoembolization (TACE) is recommended as first-line therapy for unresectable HCC and is used widely in the primary treatment of intermediate-stage disease [30,31]. There are currently no recommendations for combining TACE and ICI with ICI inhibitors as a downstaging therapy due to the lack of evidence in the literature. However, this strategy has potential benefits, as seen in a few case studies where there was no disease recurrence at 6-month follow-up following neoadjuvant use of TACE with tislelizumab or camrelizumab in HCC patients before resection [32,33]. Furthermore, since yttrium-90 radioembolization (Y90RE) and TACE have been shown in a meta-analysis to have similar efficacy and safety in the treatment of unresectable HCC, Ref. [34] adding ICI as an adjunct could possibly ameliorate this strategy. 

In view of the need to explore immunotherapy as a bridging treatment before HCC transplantation, several clinical trials are underway. The results from these trials will provide the information needed to optimize outcomes in these patients, including identifying patients who would benefit most from this strategy, determining the optimal length of ICI treatment pretransplantation, and the minimum safe washout period for different ICIs. Table 1 summarizes the ongoing clinical trials investigating immunotherapy as a bridging therapy prior to the liver transplantation. Most of these studies are ongoing, and results are expected to be available in the next few years. To the best of our knowledge, at the time of this paper, the first published prospective trial (NCT03817736) using immunotherapy and locoregional therapy as conversion therapy enrolled 33 patients in a single-arm, Phase 2 trial in patients with locally advanced HCC not amenable to curative treatment in Hong Kong and China [35]. At a median follow-up of 17.2 months, 18 (55%) of patients were amenable to curative treatment. Of these, 4 had curative treatment (resection or radiofrequency ablation), and 14 had a radiologic complete response and opted for close surveillance. Eleven (33%) of the whole cohort had Grade 3 or higher treatment-related adverse events. The most common Grade 3 or higher treatment-related adverse events were transient increases in alanine aminotransferase or aspartate aminotransferase in five patients after TACE; five patients had immune-related adverse events (two had dermatitis and three had hepatitis). A recent abstract assessing the safety and efficacy study of pembrolizumab in combination with lenvatinib in participants with HCC before liver transplant reported that early results are promising [36]. The results from these studies offer a glimpse into the potential of immunotherapy in pretransplantation HCC management.

## 3. Outcomes in Post-Transplantation Immunotherapy: Immunotherapy after Transplant

Traditionally, sorafenib has been used in the post-transplant setting with evident mortality benefit [37], with radiation therapy and localized ablation as adjunctive treatments [38]. However, tumor recurrence continues to occur in nearly 10–20% of patients post-transplantation [39], with a median survival following recurrence of approximately one year [14,40,41,42]. The risk for tumor recurrence is dependent on factors related to the tumor, the patient, or the treatment, as well as post-transplantation factors, tumor differentiation, and microvascular invasion [43]. Although immunosuppression is essential for preventing allograft rejection, it also compounds the overall risk of developing malignancy post-transplantation [44]. Due to this delicate interplay, the decision for the use of immunotherapy in liver transplant patients becomes complicated. However, with the success of immunotherapy in the management of advanced HCC, there is great interest in exploring this strategy in the post-transplantation setting.

Similar to the pretransplantation situation, there is a lack of studies evaluating the safety and efficacy of post-transplantation use of immunotherapy. Registry trials that led to the approval of ICI use in HCC excluded liver transplant recipients. Likewise, most of the data on immunotherapy post-transplantation are from case reports, case series, or single-center experiences. A review by Lominadze et al. provided a summary of the case reports/series of ICIs in the post-transplant setting [45]. Notably, a literature review by Au and Chok of case reports of liver transplant recipients who received immunotherapy found that out of 19 patients with recurrent HCC, 14 had been treated with nivolumab and five with pembrolizumab [46]. The overall objective response rate was 11%, with a median progression-free survival of 2.5 ± 1.0 months and a median overall survival of 7.3 ± 2.7 months after immunotherapy. Acute rejection occurred in 32%, and most of the early mortalities, which developed in 21% of patients, were related to acute rejection (18%). Patients with acute rejection were more likely to suffer from early mortality (56% vs. 6%). Patients who were given immunotherapy later after transplantation had a lower risk of rejection compared with patients with recent liver transplants (2.9 vs. 5.3 years). Overall, this analysis demonstrated that immunotherapy post-transplantation could be associated with fatal graft rejection, a high rate of organ failure, and early mortality. A Mayo Clinic retrospective pilot evaluation of efficacy and safety ICI in metastatic cancer patients with a history of liver transplantation found that none of the five HCC patients had a clinical benefit from PD-1 inhibition. However, this could be explained by the short duration of therapy, differences in the efficacy of the ICIs used, and the small cohort size. Only one of the five patients had graft rejection [47]. On the other hand, a systematic review by Ziogas et al. of the outcomes of patients with HCC treated with ICIs found that 3 of 14 patients (21.4%) who received ICIs in the post-transplant recurrence setting were still alive with a functional graft at 29, 20, and 10 months of follow-up after ICI initiation, respectively. Out of the 14 patients who received ICI for post-transplant recurrence, fatal graft rejection occurred in 36% and mortality in 73% [48]. Table 2 summarizes the ongoing clinical trials investigating immunotherapy in the post-transplantation setting.

## 4. Exposure to Immunotherapy and Adverse Events in Pre- and Post-Transplant Settings

### 4.1. Risk of Graft Rejection

The safety of bridging immunotherapy in both pre- and post-transplant settings appears to be a concern due to several reports of severe rejection leading to graft failure. Graft rejection in transplant patients who have undergone bridging therapy is thought to be induced by the activation of the innate immune response [23,28]. Anti-PD(L)1-based ICIs such as nivolumab and pembrolizumab stimulate the immune system by impeding the interaction between programmed cell death protein 1 (PD-1) and PD-L1. This results in downstream T cell activation, which increases the risk of graft rejection [49]. Cytotoxic T-lymphocyte–associated antigen 4 (CTLA-4)-based ICIs such as ipilimumab lead to a block between CTLA-4 and its ligands, leading to sustained activation of T cells, which induces graft rejection [50].

Recent data allude to the theory that graft rejection can be mitigated by having an adequate washout period, modalities such as plasmapheresis, and immunosuppression after transplant. In a case series of five patients, three who received the last dose of ICI > 3 months before transplant had excellent graft function with no episodes of graft rejection or HCC recurrence. Two patients who had <3 months between the last ICI dose and transplant developed severe post-transplant complications, including hepatic necrosis and graft loss, one of whom required a retransplant, which was successful [51]. In the recent retrospective review from China of patients receiving PD1 inhibitors before liver transplantation, acute rejection occurred in 9 out of 16 patients. All rejection reactions were reversed after the immunosuppression regimens were adjusted, and there was no immune-related graft loss or fatal rejection. The interval between the last PD1 inhibitor dose and transplantation was shorter in the group that experienced rejection (median 21 days) vs. the group that did not (median 60 days), a difference that was statistically significant (*p* = 0.01) [21]. In several other studies, acute rejection occurred when ICI therapy was withdrawn shortly before liver transplantation (7–16 days) [23,25,26]. These findings suggest that the time from the last ICI dose to liver transplant is an important factor in the use of ICI in liver transplantation and indicate the importance of having an adequate washout period. The half-lives of ICIs are relatively long and can last up to 4 weeks, while the pharmacological target occupancy can last even longer [52]; thus, it is important to ensure that washout periods are not short. Conversely, Tabrizian et al. reported that of nine patients undergoing nivolumab pretransplant treatment, eight received their last dose only 4 weeks before transplant. Despite this, all eight successfully underwent transplantations, with none experiencing severe allograft rejection or loss, tumor recurrence, or death [18]. The large amounts of blood transfusion required during transplantation due to significant blood loss may have led to rapid clearance of serum nivolumab, suggesting the prospect of utilizing modalities such as plasmapheresis to accelerate washout. 

In the post-transplantation setting, the occurrence of acute rejection is much lower when ICI treatment is started later, as shown in studies with longer median intervals of between 2 and 8 years after transplant [47]. For example, Pandey and Cohen reported a single patient experience of treatment of recurrent HCC 6 years after liver transplantation with ipilimumab. Although the patient experienced transient Grade 2 liver enzyme elevation, no other immune-related adverse events occurred, and the recurrent HCC resolved [53]. The risk of rejection appears to be higher when used in the early post-transplant period, as seen in a patient who experienced graft loss when she received ipilimumab 18 months after transplantation [54]. Interestingly, Munker et al. found that PD-1 expression may be linked to the risk of rejection after ICI treatment in post-transplant patients, as evidenced by the higher levels of PD-1 expression in liver biopsies with acute rejection compared with those without rejection [55].

### 4.2. Other Adverse Events 

Other adverse events should be considered in liver transplant recipients, including venous and arterial thrombosis [56,57]. ICI-induced injury in the allograft may also occur due to the altered immunologic changes associated with liver transplant and the need for chronic immunosuppression post-transplantation [58]. Anugwom and Leventhal reported a case of severe cholestatic disease in the allograft after the nivolumab treatment of recurrent HCC during the post-transplant period. The patient died from complications related to hepatic necrosis [59]. Additionally, people with pre-existing autoimmune diseases may be at higher risk of flares/exacerbations or immune-related adverse events when using ICI inhibitors [60]. For instance, a recent meta-analysis on patients with inflammatory bowel disease (IBD) and cancer found that almost 40% experienced IBD relapse during ICI inhibitor treatment, with CTLA-4 inhibitor use being associated with a higher risk. Nevertheless, the majority of the relapses were successfully managed with corticosteroids or biologic therapy, and the rates of complications and abdominal surgery were low [61].

## 5. Considerations for the Use of ICIs in Transplant Oncology

There are currently no consensus guidelines for the use of ICIs in the treatment of HCC in liver transplantation. However, data from published studies provide some guidance. In the pretransplantation setting, the timing of ICI washout is important. This is often loosely based on the serum half-life of ICIs, although the occupancy of the ICI pharmacological target on receptors should also be considered [62]. In some patients, PD-1 occupancy has been shown to be greater than 50% after 200 days following multiple doses [63]. Modalities such as plasmapheresis can be employed to speed up washout if necessary. Timing of ICI use is also important in the post-transplantation setting, where starting ICIs in the early years after liver transplantation should be approached with caution [47,54]. The choice of the agent or combinations of agents is also crucial and should be driven by the data available with regard to safety, efficacy, and response. PD-1 expression should ideally be evaluated before initiation via a liberal biopsy in post-transplantation patients. This is because PD-1 overexpression may be linked to an increased risk of rejection with PD-1 inhibitor use and may prompt the use of anti-CTLA-4 therapy instead [55]. Preliminary data suggest that ICI monotherapy may be associated with a higher rate of transplant rejection compared with combination therapies; however, this has not been specifically explored in the HCC setting [64]. The choice of immunosuppression after liver transplantation and the need for regimen adjustment before ICI initiation should also be considered, although there are few data on the impact of immunosuppression on immunotherapy response. Finally, patient expectations and preferences before starting ICI therapy in both pre- and post-transplantation settings should be considered. Patients should make an informed decision based on the efficacy and risks of adverse events, including the risk of acute cellular rejection and the potential for graft failure. 

As suggested by Ben Khaled, having an international registry that collects evidence from single-case experiences of immunotherapy in the transplant setting could help to guide future clinical studies and guidelines for use [65].

## 6. Conclusions and Future Directions

Although it is not yet entirely obvious how immunotherapy can complement transplants in the setting of advanced HCC, there is some evidence to suggest its utility, especially in the pretransplantation setting. In fact, recent United Network for Organ Sharing policy updates acknowledge the available data and now make provisions for liver transplantations for patients who have been bridged using ICI therapy. At the moment, we believe that ICI is a viable adjunct for transplant patients. With ICIs being used more frequently pre- and post-transplantation, results from randomized clinical trials specifically assessing its utility will help to clearly define parameters that enable a durable clinical response while avoiding rejection and identify patients who will benefit most from ICI treatment. This will ultimately pave the way for the development of a clinical care path for transplant patients in this setting.

## Figures and Tables

**Table 1 cancers-15-05115-t001:** Current clinical trials in immunotherapy for pretransplantation HCC.

Trial	Study Type	No. of Participants	Patient Population	Agent(s)	Primary Endpoint(s)	Status
NCT05171335; Neoadjuvant combination therapy of lenvatinib plus transcatheter arterial chemoembolization (TACE) for transplant-eligible patients with large hepatocellular carcinoma	Nonrandomized, single-arm, open-label interventional study	50	Transplant-eligible patients with HCC beyond Milan criteria	Lenvatinib	Percent tumor necrosis	Recruiting Estimated primary completion: June 2026
NCT05185505; Atezolizumab and bevacizumab before liver transplantation for patients with hepatocellular carcinoma beyond Milan criteria	Nonrandomized, single-arm, open-label interventional study	24	Transplant-eligible patients with HCC beyond Milan criteria	Atezolizumab + bevacizumab	Proportion of participants receiving liver transplant experiencing acute rejection (within 1 year after liver transplant)	Recruiting Estimated primary completion date: April 2027
NCT05475613; A prospective, single-arm study of downstaging protocol containing immunotherapy for HCC beyond the Milan Criteria before liver transplantation	Nonrandomized, Phase II, single-arm, open-label prospective study	59	Transplant-eligible patients with HCC beyond Milan criteria	PD-1 inhibitor + other targeted therapies	2-year event-free survival rate	Recruiting Estimated primary completion date: August 2027
NCT05027425; Durvalumab (MEDI4736) and tremelimumab for hepatocellular carcinoma in patients listed for a liver transplant	Single-arm, Phase II, open-label multicenter clinical trial	30	Transplant-eligible patients who have cirrhosis or portal hypertension	Durvalumab + tremelimumab	Cellular rejection rates (up to 30 days post-transplant)	Recruiting Estimated primary completion date: December 2025
NCT04425226; Safety and efficacy study of Pembrolizumab in combination with LENvatinib in participants with hepatocellular carcinoma before liver transplant as neoadjuvant TherapY—PLENTY (PLENTY202001)	Randomized, open-label clinical trial	192	Transplant-eligible patients with HCC beyond Milan criteria	Pembrolizumab + lenvatinib	Recurrence-free survival (up to ~4 years)	Recruiting Estimated primary completion date: December 2022
NCT03817736; Sequential TransArterial chemoembolization and stereotactic RadioTherapy followed by ImmunoTherapy for downstaging hepatocellular carcinoma for hepatectomy (START-FIT)	Nonrandomized, Phase II, single arm open-label interventional study	33	Advanced HCC	ICI (not stated)	Number of patients amendable to curative surgical interventions (resection or transplantation after successful downsizing of tumor(s) with intervention; ~3 years)	Recruiting Actual primary completion date: 14 June 2022 Estimated study completion date: January 2023
NCT04443322; Safety and efficacy study of durvalumab in combination with lenvatinib in participants with locally advanced and metastatic hepatocellular carcinoma—DULECT2020-1 trial	Nonrandomized, single-arm, open-label interventional study	20	Locally advanced HCC before liver transplant and metastatic HCC	Durvalumab + Lenvatinib	Progression-free survival (up to 3 years) Recurrence-free survival (up to 4 years)	Recruiting Estimated primary completion date: December 2021
NCT05879328; Liver transplantation in patients with partial or complete response after atezolizumab plus bevacizumab for intermediate-advanced stage hepatocellular carcinoma: the ImmunoXXL Study	Prospective, single-arm observational study	12	Patients with HCC beyond transplant criteria who had undergone liver transplantation after downstaging	Atezolizumab + bevacizumab	Recurrence-free survival (up to 2 years)	Recruiting Estimated primary completion date: December 2024
NCT04814043; Systemic PD-1 antibody (sintilimab) and lenvatinib plus transarterial chemoembolization and FOLFOX-based chemotherapy infusion for potential resectable HCC: a single-arm, Phase 2 clinical trial	Nonrandomized, single-arm, Phase 2, open-label interventional study	57	Patients with potentially resectable HCC	Sintilimab + lenvatinib	12-month conversion rate to resection	Recruiting Estimated primary completion date: December 2022

HCC, hepatocellular carcinoma; PD-1, programmed cell death protein 1; PD-L1, programmed death-ligand 1.

**Table 2 cancers-15-05115-t002:** Current clinical trials in immunotherapy for post-transplantation HCC.

Trial	Study Type	No. of Participants	Patient Population	Agent(s)	Primary Endpoint(s)	Status
NCT05411926; Effect of PD-1/PD-L1 inhibitor therapy before liver transplantation on acute rejection after liver transplantation in patients with hepatocellular carcinoma	Single-center, prospective, noninterventional cohort study based on real-world data	30 cases 30 controls	Patients with HCC who had undergone allogenic liver transplantation with/without prior PD-1/PD-L1 monotherapy	PD-1/PD-L1 inhibitor monotherapy	Incidence and severity of acute rejection, cellular immune function after liver transplantation. Dose and drug concentration of tacrolimus after liver transplantation.	Recruiting Estimated primary completion date: March 2023
NCT05913583; Correlation between exposure to immune checkpoint inhibitors before liver transplantation for hepatocellular carcinoma and post-transplant graft rejection	Retrospective, observational study	160	Patients with HCC who had undergone liver transplantation	ICIs (not specified)	Graft rejection within 1 year after liver transplantation	Recruiting Estimated primary completion date: September 2023

HCC, hepatocellular carcinoma; ICIs, immune checkpoint inhibitors; PD-1, programmed cell death protein 1; PD-L1, programmed death-ligand 1.

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
