# Peer review of "Hepatocellular Carcinoma: The Role of Immunotherapy and Transplantation in the Era of Transplant Oncology"

_cancers, 2023, doi:10.3390/cancers15215115_

Round 1

Reviewer 1 Report

This is a review article on the role of immunotherapy in liver transplantation for hepatocellular carcinoma.

I have some comments.

1.        Put the line number for the reviewer’s convenience.

2.        (the 1st page, Line 4 from the bottom) Spell out “FDA”.

3.        (the 2nd page, Line 9) Spell out “ECOG”.

4.        (the 2nd page, Line 9 from the bottom) Spell out “PD-L1”.

5.        (the 3rd page, Line 2) What are mRECIST criteria?

6.        (the 3rd page, Line 14) What is OLT?

7.        (the 3rd page, Line 21) What is AFP?

8.        (the 4th page, Line 2) Spell out “TRAEs”.

9.        (the 4th page, Line 3) Replace “TACE” with “transcatheter arterial chemoembolization”.

10.     (the 4th page, Line 4) What is (HCC)?

11.     (the 6th page, Line 8 from the bottom) What is CTL?

12.     (the 7th page, Line 26 from the bottom) What is AEs?

13.     (the 8th page, Line 3) What is CTLA-4? What is the difference with CTL-4 in the page 6?

The English is good, needed only small revisions.

Author Response

Reviewer 1:

We made all suggestions regarding spelling out the abbreviations.

  1. (the 8th page, Line 3) What is CTLA-4? What is the difference with CTL-4 in the page 6?

We have included the full form of CTLA-4 and standardized all to CTLA-4. They were referring to the same thing.

Reviewer 2 Report

In this report, the authors review the status of the use of immune checkpoint inhibitors (ICIs) as a potential alternative to be in those patients affected with hepatocellular carcinoma (HCC) which may undergo to a liver transplant. Authors reviewed the  actual criteria systems used to consider a oncologic patient to liver transplantation (Milan and UCSF criteria), and the use of immunotherapy to downstaging patients affected with advanced oncologic disease to be acceptable to liver transplant. They review the use of ICIs before transplant and review the actual clinical trials which are currently ongoing. The also review the use of immunotherapy after to control HCC progression after transplantation and finally they review the adverse events which the use of ICIs may induce in patients either in pre- and post-transplant settings. The conclude that the currently evidence suggest that the f ICIs may be useful as a complementary therapy for advance HCC especially in pre-transplantation setting, and the clinical trials which are ongoing may be helpful to define which patients will benefit most of this therapy.

In my opinion, the topic is of clinical interest and original. The paper is well written, is easy to follow the topics and use a clear language, even to those readers which are non-specialist in the field. References are updated, and there is no self-citations or signs of plagiarism. Therefore, I would recommend this paper to be published in this journal. My only minor point is about the tables, which should be better organized (Table 2 seems to loose the heads). Additionally there are some typographical mistakes (I-E. Page 4, line 6, or Table 1 trial NCT04425226, LENvatinib), so a revision of the full text is recommended.

Author Response

Reviewer 2

Typographical mistakes e.g. LENvatinib

We have left a note to you for explanation in the manuscript.  

Reviewer 3 Report

Very interesting and well written review on a very specific topic. 

Some figures would improve the quality of the manuscript.

I suggest to add a comment on the safety issues with immunotherapy, with particular reference to the risk of flare of pre-existing immune diseases (cite the recent MA: PMID: 33314269)

Could the authors add some comments on the potential role of immunotherapy combined with loco-regional therapies as bridging/downstaging therapy in the pre-transplant setting? (on this regard, cite the MA: PMID: 27366304)

Author Response

Reviewer 3

IO and pre-existing immune diseases -We added a couple of lines and included the recommended citation.

IO and locoregional therapies - As above, added a few lines and included the recommended citation.

Round 2

Reviewer 3 Report

The revised version of the paper is OK. Thank you!